

# Biogeochemical contrast between different latitudes and the effect of human activity on spatio-temporal carbon cycle change in Asian river systems

Tadanobu Nakayama[1]

[1]Center for Global Environmental Research, National Institute for Environmental Studies, Tsukuba, 305-8506, Japan

*Correspondence to*: Tadanobu Nakayama (nakat@nies.go.jp)

**Abstract.** Recent research has shown inland water may play some role in carbon cycling, although the extent of its contribution has remained uncertain due to limited amount of reliable data available. In this study, the author applied an advanced model coupling eco-hydrology and biogeochemical cycle (NICE-BGC) to regional-continental scales, which incorporates complex coupling of hydrologic-carbon cycle and interplay between inorganic and organic carbon. The author evaluates latitudinal effect and human impact on hydrologic and carbon cycles between boreal Ob River, temperate Yangtze River, and subtropical Mekong River basins in Asia by using different resolutions of river network data. The model simulated more heterogenous distributions of water and carbon flux in the finer river network data in these regions, and helped to identify some hot spots on a regional scale. Then, the model was extended to continental scale at 1°x1° resolution with a time step of $\Delta t = 1$ day to evaluate seasonal and diurnal variations in carbon flux parameters. The model result showed there is a seasonal variability of horizontal transport and vertical fluxes among boreal, temperate, and tropical regions and among each continent, which reflects seasonal variations of biologic and hydrologic processes there. The result showed $CO_2$ evasion increases and sediment storage decreases in nighttime, particularly clearly seen temporarily in summer in Yangtze River, which implied some hot spots and hot moments in the day-night difference of vertical fluxes in regional scale. These results emphasize the important role of Asian river systems on global carbon cycle, and the further need to improve the resolution of simulation, to implement carbon observation network, and to apply satellite data in the higher-resolution.

## 1 Introduction

Previous researches have suggested that variability and uncertainty in carbon cycle in terrestrial ecosystem are relatively larger than those in atmosphere and ocean spatio-temporally (Le Quéré et al., 2009), and that the terrestrial biosphere sequesters most available carbon there. Recently, some research has started to reconsider the importance of inland waters including rivers, lakes, and groundwater different from terrestrial ecosystem (Cole et al., 2007; Battin et al., 2009). In particular, inland waters play a role in transport, mineralization, and sequestration of carbon, which also might be complicated by surface−groundwater interactions around wetland and riparian areas; water movement and the temperature there drive carbon storage and flux complicatedly.



The author has so far developed the process-based National Integrated Catchment-based Eco-hydrology (NICE) model (Nakayama, 2014, 2015), which includes complex interactions between the forest canopy, surface water, the unsaturated zone, aquifers, lakes, and rivers. The model can simulate iteratively nonlinear interactions between hydrologic, geomorphic, and

ecological processes, and include feedback and down-scaling process from regional simulation to local simulation with finer resolution (Nakayama, 2014). The author previously attempted to extract the impacts of groundwater-level change, sediment deposition, and nutrient availability on the complex vegetation succession process, and vice versa by using NICE. Recent research suggests serious concerns against extrapolation of experimental results at a small scale to entire landscapes and therefore inevitably the necessity to bridge the gap between ecosystems at various scales (Deegan et al., 2012). It is powerful

to re-evaluate the ecosystem as an extension of the "metabolic theory of ecology" (Brown et al., 2004) from the perspective of a meta-ecosystem analysis by considering multi-scaled aspects between global–regional–micro levels in the same way as the "river continuum concept" (Vannote et al., 1980). Hotchkiss et al. (2015) further evaluated the relative role of rivers as conduits for terrestrial $CO_2$ efflux and as reactors mineralizing terrestrial organic carbon as a function of the size and connectivity with landscape in United States. From this point of view, the author's previous improvement in the model (Nakayama, 2014) helps

to bridge the gap of carbon cycle between ecosystems at various scales.

About the carbon cycle in inland waters, previous researches evaluate biogeochemical cycle in inland water to indicate that carbon budgets are diverse at various basins/catchments and that there is a close and complex relationship between $pCO_2$ (partial pressure of $CO_2$ in water), DOC (Dissolved Organic Carbon), DIC (Dissolved Inorganic Carbon), and POC (Particulate

Organic Carbon) in inland water (Cole et al., 2007; Tranvik et al., 2009; Aufdenkampe et al., 2011; Raymond et al., 2013; Nakayama, 2016). While $pCO_2$ and $pCH_4$ are necessary to evaluate $CO_2$ and $CH_4$ flux to the atmosphere (evasion), DOC, DIC, and POC are also important to evaluate $CO_2$ flux to the ocean and sediment storage. As for a temporal scale, the diurnal variation of carbon flux remains poorly evaluated except some researches (Peter et al., 2014), which pointed out that $CO_2$ outgassing fluxes during nighttime exceeds those during daytime. From this viewpoint, it is necessary to further develop a

process-oriented model to obtain a better grasp of the biogeochemical cycle in the biosphere (terrestrial and aquatic ecosystems) as 'unifying currency' through a holistic approach (Prairie and Cole, 2009). This would help to clarify the mechanism of the carbon cycle in more detail, particularly the spatio-temporal interplay between inorganic and organic carbon and its relationship to nitrogen and phosphorus, etc. in the context of the hydrologic cycle.

NICE was further developed to couple it with five biogeochemical cycle models, including those for terrestrial ecosystems, those for water quality in aquatic ecosystems, and those for carbon weathering. The revised NICE model (NICE-BGC) incorporates the connectivity of carbon, nitrogen, and phosphorus cycles accompanied by the hydrologic cycle between surface water and groundwater, hillslopes and river networks, and other intermediate regions (Nakayama, 2016, 2017a). The model simulates both horizontal transport to the ocean and vertical fluxes and includes aquatic metabolism and terrestrially derived




carbon together in major rivers, which is a potential improvement from the previous studies (Cole et al., 2007; Tranvik et al., 2009; Aufdenkampe et al., 2011; Raymond et al., 2013).

5 There are two hypotheses in case of applying this new model as follows; (i) Latitudinal difference is an important factor to constrain a variability of horizontal and vertical carbon fluxes, and (ii) Seasonal and diurnal variations are important components to evaluate the impact of regional hot spots on global carbon cycle change. Based on these hypotheses, in the present paper, NICE-BGC was applied to Ob River basin including West Siberian Lowland (WSL) (approximately 47 to 67 degrees north), Yangtze River (approximately 25 to 35 degrees north), and Mekong River basin (approximately 10 to 34 degrees north) in Asia for the evaluation of latitudinal effect and human impact on eco-hydrological process and 10 biogeochemical cycles (Fig. 1). The model was applied to these three rivers representative of boreal, temperate, and tropical/subtropical regions for the evaluation of diurnal carbon cycle in the downstream of the rivers. Further, the model was extended to six river basins to differentiate seasonal variations of carbon budgets there. Finally, the model was extended to continental scales to evaluate eco-hydrological cycle and the role of Asian river systems in the seasonal and diurnal biogeochemical cycle change. The new model could show that there is a great variability of DOC, POC, and DIC horizontal 15 transport to the ocean reflecting biologic and hydrologic processes, and vertical flux of $CO_2$ degassing affected by both terrestrially derived $CO_2$ and $CO_2$ production through aquatic metabolism, which were usually evaluated separately in the previous researches in regional and continental scales, and so there is a great improvement from them (Nakayama, 2016, 2017a).

## 2 Regional Overview of Ob River, Yangtze River, and Mekong River Basins Representative of Asian River Systems

20 The latitudinal difference of carbon cycle in basins is assumed as an important factor to constrain a variability of horizontal and vertical carbon fluxes. Based on this hypothesis, the present paper firstly chooses three basins with different latitudes as a representative of Asian carbon cycle, and then extended to continental scale.

Ob River includes WSL, which contains the world's most extensive peatlands (Smith et al., 2004). In particular, boreal and 25 subarctic peatlands in such as WSL store about 15-30% of the world's soil carbon as sink (Limpens et al., 2008), and affect the dynamics of methane emission as source. Wetlands have also important influences on hydrologic cycle, provides important role on biogeochemical process, and helps preserve valuable species. The WSL is located on the lower Ob River basin of permafrost transition (Fig. 1). So, the lower Ob, frequently inundated during spring and summer, is different from the middle and the upper Ob in terms of hydrological regime and watershed management. It is important to evaluate eco-hydrological 30 processes there by applying advanced process-based model from the view point of biogeochemical cycle.



In the Yangtze River in humid south (Fig. 1), deforestation and land reclamation induce serious soil erosion and increase floods, in particular, shrinkage of flood storage ability around lakes is serious (Nakayama and Watanabe, 2008; Nakayama and Shankman, 2013). Although Three Gorges Dam (TGD) provides flood control and other benefits, changes in the aquatic environment of the estuary and the East China Sea might be brought about by changes in pollutant loads caused by the

deposition of large amounts of sediment carried from the upper region in the dam and the artificial control of discharge volume (Yang et al., 2006). Some recent researches have pointed out that the impact of TGD on flood occurrence in Changjiang downstream has also increased problem against the original justifications for building the dam (Nakayama and Watanabe, 2008; Nakayama and Shankman, 2013).

In contrast, Mekong River is originated in the Himalayas and the longest river in the Southeast Asia (Fig. 1). Supersaturated $pCO_2$ conditions are also typical in this subtropical river, and suspended sediment loads are also dynamic in the same way as Amazon River (Alin et al., 2011). Though there are no dams on the lower mainstream, many dams are planned or under construction for hydropower in the river system. These will transform the original river by altering natural flow patterns and disrupting fisheries and other ecosystem services soon (Grumbine et al., 2012).

**3 Materials and Methods**

**3.1 Process-based NICE model**

The original NICE (National Integrated Catchment-based Eco-hydrology) framework includes the following complex sub-compartments (Nakayama, 2014, 2015): surface hydrology submodel, land-surface submodel including urban and crop processes; groundwater submodel; regional atmospheric submodel; mass transport submodel for constituents in sediment and

water; and vegetation succession submodel. The model also includes surface–groundwater interactions assimilating land-surface processes to describe variations of leaf area index (LAI) and fraction of photosynthetically active radiation (FPAR) derived from satellite data. LAI and FPAR are important parameters for evaluating vegetation growth (Nakayama and Watanabe, 2004).

**3.2 Coupling NICE with biogeochemical cycle models (NICE-BGC)**

In order to fill the gap in the current eco-hydrology, Nakayama (2016, 2017a) has recently coupled the original NICE framework with various biogeochemical cycle models, including those for terrestrial ecosystems such as LPJWHyMe (Lund‐Potsdam‐Jena Wetland Hydrology and Methane) (Wania et al., 2010), those for water quality in aquatic ecosystems such as QUAL2Kw (Pelletier et al., 2006) and SWAT (Soil and Water Assessment Tool) (Neitsch et al., 2011), those for carbon weathering such as RokGeM (Rock Geochemical Model) (Colbourn et al., 2013), and CO2SYS (CO2 System Calculations)

(Lewis and Wallance, 1998) (Fig. A1 in Appendix A).



In the modified NICE model (NICE-BGC), all submodels were coupled together with the original NICE to conserve the carbon budget (Nakayama, 2016, 2017a). Each submodel offers iterative simulation in the most efficient way by combining on-line and off-line modes (on-line: data input/output through I/O memory, off-line: data input/output through file). This means that the newly coupled model incorporates the connectivity of the biogeochemical cycle accompanied by the hydrologic cycle for surface water and groundwater, hillslopes and river networks, and other intermediate regions (Fig. A1 in Appendix A) (Nakayama, 2016);

(i) LPJWhyMe is a further development of the LPJ (Lund‐Potsdam‐Jena Dynamic Global Vegetation Model) (Sitch et al., 2003), which simulates plant physiology, carbon allocation, decomposition, and hydrological fluxes. The author modified the original model (LPJWhyMe v1.3.1) to input more accurately the results for groundwater level, surface runoff, and evapotranspiration simulated by the original NICE at every time step. While the original model simulates water table position (groundwater level) by the simple box model of Granberg et al. (1999), and simulates surface runoff, evapotranspiration, photosynthesis, and gross primary production as a function of groundwater level, the author modified to simulate the surface runoff, evapotranspiration, and groundwater level by NICE.

(ii) SWAT is a river basin/watershed-scale model designed to predict the impact of land management practices on water, sediment, and agricultural chemical yields in large complex watersheds with varying soils, land use, and management conditions (Neitsch et al., 2011). The model can simulate the loadings of water, sediment, nutrients, and pesticides from the hillslope to the river channel, and then the loadings are routed through the stream network of the watershed in daily time steps. However, the original model can simulate only organic carbon and inorganic carbon, and water temperature is assumed to be a linear function of daily-averaged air temperature for a well-mixed stream in the simpler way (Stefan and Preud'homme, 1993; Raymond et al., 2013) than the heat-budget method such as in QUAL2Kw. The author used SWAT only to simulate diffuse sources of hydrologic and pollutant loads from the upland watershed and groundwater, and replaced SWAT with the enhanced stream water quality model QUAL2Kw for streams. The author also assumed that the temperature of groundwater was constant (=15 °C), and TSS, slow CBOD, fast CBOD, phytoplankton, and detritus there were all set as 0 because there is no simulation in SWAT and it is reasonable to set a constant water temperature with no DOC and POC in groundwater in the first approximation.

(iii) RokGeM is a spatially explicit model of carbonate and silicate rock weathering with climate feedback, and simulates weathering fluxes of alkalinity and DIC dependent on, and in feedback with, inputs of soil temperature ($T$), runoff ($R$), and plant productivity ($P$) (Colbourn et al., 2013). The author modified the original model (RokGeM v0.9) to input more accurate results for soil temperature and runoff simulated by the NICE, and for NPP (net primary production) simulated by the revised LPJWhyMe at every time step because of the coarse nature of the original one. The author used a 2-D scheme based on GKWM (Gibbs and Kump Weathering Model) (Gibbs and Kump, 1994) to calculate fluxes of calcium ions from carbonate and silicate weathering, which applies an empirical linear relationship between the fluxes of atmospheric/soil $CO_2$ from chemical





weathering and runoff. This modification improved the accuracy of the simulated weathering fluxes of alkalinity and DIC, which were input to the revised CO2SYS simulation as described in the following.

(iv) CO2SYS is a module for calculating parameters related to the $CO_2$ system in seawater and freshwater, and uses two of the four measurable parameters of the $CO_2$ system (total alkalinity, total inorganic $CO_2$, pH, and either fugacity or partial pressure

of $CO_2$) to calculate the other two parameters under equilibrium condition for a set of input conditions (temperature and pressure) and a set of output conditions chosen by the user (Lewis and Wallance, 1998). The author modified the original model (CO2SYS v2.1) to calculate automatically the time-series of the other two parameters in global rivers for 2 years during 1998-1999 using Microsoft Visual Basic 2013. In the NICE-BGC, the revised LPJWHyMe simulates DIC, and $pCO_2$ is assumed to be a relatively high constant value (=10,000 μatm) in the shallow groundwater of wetlands because of the high

$CO_2$ supersaturation there (the soil type is organic) (Cole et al., 2007); these are automatically converted to alkalinity and pH using the revised CO2SYS. The revised RokGeM simulates alkalinity and DIC in relatively deep groundwater of other terrestrial ecosystems (where the soil type is not organic), and these are also automatically converted to $pCO_2$ and pH using the revised CO2SYS.

(v) QUAL2Kw is a river and stream water quality model that simulates the transport and fate of a number of constituents such

as temperature, carbonaceous biochemical oxygen demand, dissolved oxygen, phytoplankton and several forms of the nutrients phosphorus and nitrogen (Pelletier et al., 2006). The model simulates water temperature, pH, alkalinity, inorganic suspended solids, pathogenic bacteria, and bottom algae by solving a set of partial differential equations. In this paper, the original QUAL2Kw (qual2kw60b06a01.xlsm) was modified to input automatically a times-series of lateral sources from hillslope runoff and groundwater simulated by the revised versions of LPJWHyMe, SWAT, RokGeM, and CO2SYS in each channel

because of the need to input manually a time-series of lateral sources in the original model before the simulation started. The simulated results for a channel in the first step were input to the downstream simulation as upstream conditions in the second step, and the revised model could simulate automatically all the channels, from the uppermost to the lowest, by reading all the information for each channel (location, length, slope, width, roughness, shape, etc.) step by step for longer simulation periods. Table 1 shows a list of input parameters from terrestrial ecosystems into the revised QUAL2Kw. The author assumed some

parameters were constant, and the units in input values were converted for input to the revised QUAL2Kw simulation.

The details are described in the author's recent paper (Nakayama, 2017a, 2017b). The above modifications helped to simulate a huge number of global rivers automatically in one simulation and enabled the NICE-BGC to be a process-oriented biogeochemical cycle model of human-nature-coupled phenomena; mechanisms of transport, mineralization, and sequestration

of carbon in terrestrial-aquatic linkages, interplay between inorganic and organic carbon (DOC, POC, DIC, $pCO_2$, etc.) and its relationship to nutrients (e.g., nitrogen and phosphorus, etc.), chemical weathering, food chains in aquatic ecosystems (photosynthesis and respiration of phytoplankton and bottom algae, and detritus, etc.), degassing of supersaturated $CO_2$ particularly in lakes and the upstream stretches of rivers, $CH_4$ emission to the atmosphere sensitively affected by groundwater level in wetlands and paddy fields, fertilizer application to agricultural fields of about 20 crops, pollutant loading from urban



areas, $CO_2$ and $CH_4$ emissions from reservoirs and bottom sediments, and DOC, POC, and DIC efflux to the ocean, etc. In particular, the model in this study can simulate the diurnal patterns (sub-day scale) of water and carbon cycles in rivers (photosynthesis in daytime and respiration in nighttime) complicatedly affected by the above components. This new process-based model also enables to separate more clearly the carbon sources of inland waters from the terrestrial carbon sink (Nakayama, 2016), whereas most previous research on conventional carbon cycling has generally overestimated the accumulation of carbon in terrestrial ecosystems and underestimated the potential for lateral export of carbon from land to rivers (Raupach, 2011).

### 3.3 Model input data

Six-hour re-analyzed data for downward radiation, precipitation, atmospheric pressure, air temperature, air humidity, wind speed, cloud cover, FPAR, and LAI were input to the original NICE after interpolation of the ECMWF-Reanalysis Interim Product (ERA-interim) (ECMWF, 2013) in inverse proportion to the distance back-calculated in each grid. These meteorological data were used to calculate daily-averaged values, and then input to NICE-BGC in addition to soil temperature, surface runoff, and groundwater level simulated by NICE. The input data for the global simulation were prepared and arranged by using ArcGIS v10.1 software; elevation, land cover, soil texture, vegetation type, river networks, lakes and wetlands, and geological structures were categorized on the basis of the global digital elevation model (DEM; GTOPO30) (U.S. Geological Survey, 1996a), Global land Cover 2000 (GLC2000) (European Environment Agency, 2015), Harmonized World Soil Database (HWSD) (European Commission, 2012), GLDAS Vegetation Class (NASA, 2013), HYDRO1K (U.S. Geological Survey, 1996b), Global Lakes and Wetlands Database (GLWD) (Lehner and Döll, 2004), and Global Lithological Map Dataset (GLiM) (Hartmann and Moosdorf, 2012) in addition to GKWM (Gibbs and Kump, 1994), respectively. About regional simulation of Ob River and Mekong River basins, the author used three resolutions of river data for each simulation; 12 stream channels and 128 stream channels by HYDRO1K (U.S. Geological Survey, 1996b), and 1160 stream channels by GRDC (GRDC, 2014) in Ob River basin, and 19, 82, and 449 stream channels by HYDRO1K (U.S. Geological Survey, 1996b) in Mekong River basin, respectively (Fig. 2, and Table 2). The author used only one type of river data for simulation; 9 stream channels by HYDRO1K (U.S. Geological Survey, 1996b) in Yangtze River basin in comparison with the author's previous simulation of detailed hydrologic cycle (Nakayama and Watanabe, 2008; Nakayama and Shankman, 2013).

The author further subdivided the agricultural regions of GLC2000 into 17 major crops of 26 crop classes for both irrigated and rain-fed crops in MIRCA2000 (Portmann et al., 2010), corresponding to the crop commodity classifications and definitions reported previously (Leff et al., 2004). The author also applied Global Map of Irrigation Areas (GMIA) (FAO, 2016) to decide the irrigation type (surface water, groundwater, and other), and Global Crop Water Model (GCWM) (Siebert and Döll, 2010) to decide irrigation water use. Total fertilizer consumptions (nitrogen, phosphorus, and potassium) for Earth Stat (Mueller et al., 2012) were used to calculate the average fertilizer consumptions for each crop. These data are important to assess human impacts on the Asian river systems where the human activity has dramatically changed hydrologic cycle, biogeochemical cycle,





and ecosystem dynamics (Nakayama, 2015). Furthermore, the authors applied a crude rule whereby the anisotropy ratio and hydraulic conductivity were based on soil texture class, and the general relationship whereby permeability decreases exponentially with the ratio of depth to e-folding length (a function of the slope), as described by Fan et al. (2013).

### 3.4 Boundary conditions and running the simulation

At the upstream boundaries, conditions at the hydraulic head were based on an assumption of no inflow from the mountains in the opposite direction. For the hillslope hydrology, flow depth, discharge, and other water concentrations on the uppermost boundaries of basins were set to zero throughout the simulation. In order to calculate these boundaries, the author firstly used GTOPO30 to calculate each basin boundary at 1km x 1km resolution, and then calculated 1x1 degree grid by using ArcGIS v10.1 software for the model simulation. Water temperature in the uppermost boundary of each river was calculated as an

exponential function of daily-averaged air temperature (Yearsley, 2012). Inflows or outflows from the riverbeds were simulated at each time step depending on the difference in the groundwater and river hydraulic heads. At the sea boundary, a constant head was set at 0 m. Details are described in Nakayama and Watanabe (2004).

The simulation was conducted in horizontally about 2,100 km wide by 3,000 km long with resolutions of 0.15 ° and 1 ° in the

Ob River basin and in horizontally about 1,700 km wide by 2,600 km long with resolutions of 0.10 ° and 1 ° in the Mekong River basin (Figs 1 and 2), and these areas were discretized into 20 layers with a weighting factor of 1.1 layers (layer thicknesses being graduated from thinner at the upper layer to thicker at the deeper layer) in the vertical direction. The upper land layer was set at 2 m depth, and the 20th land layer was defined as an elevation of –200 m from the sea surface. So, the total thickness of the 20 layers vary from 200 m to 5,282 m in the Mekong River, and from 200 m to 1,952 m in the Ob River.

The model simulates river network among elements and terrestrial ecosystem in grid cells. Then, the contribution of river on grid cell was estimated by calculating area ratio of river (multiplication of river width and length) in each grid. Simulations were performed with a time step of $\Delta t$ = 6 h for 1998-1999 after 1 year of warm-up period until a time-varying equilibrium. The model was also extended to the global scale at 1°x1° resolution with a same time step as in WSL and Mekong River. The simulated results were calibrated and validated in hydrologic, geomorphic, and ecological aspects respectively in the author's

previous researches (Nakayama, 2014, 2015).

After that, NICE-BGC simulation for terrestrial ecosystem was conducted at the same spatial resolution with a time step of $\Delta t$ = 1 day for the same period by inputting some of the results simulated by NICE (soil temperature, surface runoff, and groundwater level) in off-line mode after calculating the daily-averaged data from 6-hourly data (Fig. A1 in Appendix A).

Then, NICE-BGC simulation for aquatic ecosystem was conducted with more smaller time step between $\Delta t$ = 0.044 min and 0.70 min for the stability of the model, which made it possible to simulate the diurnal water and carbon cycles in inland waters. In particular, spin-up in the revised RokGeM was conducted for much longer periods of 20,000 years under control conditions (no emissions) to allow complete equilibration of silicate weathering versus $CO_2$ outgassing at a pre-industrial level



(atmospheric $pCO_2$ was about 278 ppm), being much longer than the periods used in other submodels, such as 1,000 years in the revised LPJWHyMe and 6 months in the revised SWAT and QUAL2Kw. The continental values are estimated by a compilation of the major river basins in the world (153 basins/watersheds, 325 river channels) as in Nakayama (2017a). Though the model simulated only the large rivers, the hydrologic and carbon cycles were simulated within the range of previous researches in 27 different rivers (Fig. A2 in Appendix A), as described in Nakayama (2017a). The details of NICE-BGC simulation were described in the author's previous study (Nakayama, 2017a).

## 4 Results

### 4.1 Difference of hydrologic and carbon cycles between Ob River and Mekong River

Fig. 3 shows annual-averaged water and carbon flux from terrestrial into aquatic ecosystems in three types of river data (Fig. 1) simulated by NICE-BGC. The simulated results were compared in different latitudes of Ob River and Mekong River. Simulated results show the model simulated more heterogenous distributions of water and carbon flux in the finer river network data in both regions. Though it was difficult to validate these fluxes, the author validated the water and carbon flux in major rivers in the figures as mentioned later and his previous researches as much as possible (Nakayama, 2016, 2017a, 2017b). In particular, the model shows that there is a clear difference of hydrologic and carbon cycles between these regions caused by meteorological conditions, topography, land use, soil property, and geological characteristics, etc. (Jenerette and Lal, 2005). There is more flux in the downstream region in Ob River, whereas more flux in the upper region in Mekong River. Because the surface water is more dominant than the groundwater in West Siberia (percentage of total flux to the ocean; surface water = 77.4 %, groundwater = 22.6 %), DOC flux, mainly originated from the wetland, has a great effect on TOC flux (= DOC + POC + DIC flux) (Figs 3a-b). It can be seen more clearly that DOC flux increases in the wetland from where more organic carbon is exported. POC flux increases in the downstream of Ob River where land cover changes from forest to wetland. DIC flux takes the greater value where peat depth is large, which has a great effect on the distribution of total carbon flux (Fig. A3 in Appendix A). In contrast, DIC flux takes the greater value in the basin, particularly in the upper Mekong region because the groundwater is more dominant than the surface water there (surface water = 11.2 %, groundwater = 88.8 %) (Figs 3c-d). Much rainfall effected by monsoon and rapid change in the topography from the upper to middle/lower regions causes a greater runoff in the Mekong River (Fig. A4 in Appendix A).

Table. 3 shows the comparison of annual runoff and carbon flux in three types of river data in West Siberian Lowland (including Ob River basin) and Mekong River basin. The simulated results are generally in the range of previous data (Schlunz and Schneider, 2000; Coynel et al., 2005; Alin et al., 2011; Dai et al., 2012; Long et al., 2015), as evaluated in the author's previous studies (Fig. A2 in Appendix A) (Nakayama, 2017a, 2017b). The simulated results also show runoff and carbon fluxes increase in the finer river network data in the Ob River, whereas the fluxes vary more complicatedly in the Mekong River. This is mainly because of a narrow strip of upstream region in the Mekong (Fig. 3) by using relatively large stream



order in the simulation (Table 2), and because of the larger variations of the total thickness of the vertical layers in the model there, as described in the section 3. However, the importance is that the present study investigates the sensitivity of the model to the various resolutions of the drainage network used in the model, and that the total water and carbon flux are relatively similar regardless of the different resolutions of the drainage network. It is also interesting that there is not a great difference

in the flux of upstream region in three types of river data. The difference in the flux becomes larger in downstream region, and this difference is more predominant in the stream channel than in the hillslope. The simulated result implies the effect of river network data on carbon cycle variability, which is also related to the importance of dry watercourses, intermittent rivers, and temporary waterways (von Schiller et al., 2014). The other research also indicates the importance of dry rivers that they constitute more than 30 % of the total length and discharge of the global river network (Datry et al., 2014), and that 69 % of

first-order streams below 60° latitude flow only intermittently (Raymond et al., 2013). Though the present study treats only relatively large stream order (5 – 10) according to Downing et al. (2012) (Table. 2) because of computational load as small as possible in regional scale, it is necessary to evaluate the effect of dry or intermittent watercourses on carbon cycle change in smaller stream order in future. The dry phase of intermittent rivers can be biogeochemically active and may release substantial $CO_2$ to the atmosphere, whereas large quantities of organic material and nutrients can produce hot moments of biogeochemical

transfer and transformation processes during flood seasons.

## 4.2 Diurnal and seasonal variations of carbon cycle in local scale

Fig. 4 shows the simulated result of time-series of discharge and carbon flux in Ob (Salekhard station; boreal), Yangtze (Datong station; temperate), and Mekong (Stung Treng station; tropical/subtropical) Rivers, respectively. The model generally reproduced the observed discharge (Changjiang Water Conservancy Committee, 1998, 1999; Lammers and Shiklomanov,

2000; Mekong River Commission, 2011) during 1998 – 1999 in three regions (Figs 4a-c). In particular, the correlation of discharge is higher ($R$=0.983) in Yangtze River than in Ob ($R$=0.339) and Mekong ($R$=0.791) Rivers because the original NICE was verified in detail by using observation data of river discharge, soil moisture, and groundwater level in Yangtze River basin in the author's previous research (Nakayama and Watanabe, 2008). The discharge data at Stung Treng gauging station in the downstream Mekong River (Mekong River Commission, 2011) was digitized from the figure in Lauri et al. (2012). The

model result shows that there are seasonal variations of carbon cycles, which imply the complicated interaction between inorganic and organic carbon in the river (Jenerette and Lal, 2005). In particular, there was a serious flood in 1998 summer in Yangtze River, and the model implies that this caused high variations of carbon flux during that extreme period (Reichstein et al., 2013). The model can calculate the diurnal carbon cycle, and it can be seen clearly that $CO_2$ evasion increases and sediment storage decreases in nighttime, particularly in Yangtze River (Figs 4d-f). This result suggests that temperature-induced

physico-chemical processes can't be ignored in aquatic metabolism whereas hydrologic cycle affects primarily the dominance of organic or inorganic carbons in the river as some studies pointed out (Hope et al., 1994; Leach et al., 2016). This trend is similar to Peter et al. (2014) in temperate region of Austria, which suggests that (i) enhanced outgassing would reduce $CO_2$ accumulation in the stream water and hence depress $pCO_2$ as stream water temperature increases during daytime, and that (ii)



primary production adds to the diurnal reduction of stream water $pCO_2$ besides temperature-induced physical processes. The model further clarified that these mechanisms are closely related to the slight increase in DIC and the decrease in POC in nighttime in three rivers.

### 4.3 Seasonal variation of carbon cycle in regional scale

Fig. 5 shows seasonal variations of carbon flux in six major rivers in boreal (Ob and Yukon Rivers), temperate (Yangtze and Mississippi Rivers), and tropical/subtropical (Mekong and Amazon Rivers) regions simulated by the NICE-BGC. The author has compiled as many previous materials as possible and calculated an averaged value (Nakayama, 2017a, 2017b) in order to estimate the carbon cycle both in vertical flux such as $CO_2$ evasion to the atmosphere and sediment storage, and in horizontal transport such as TOC, DOC, POC, DIC-flux to the ocean in previous data of these rivers (Schlunz and Schneider, 2000;

Coynel et al., 2005; Rasera et al., 2008; Dubois et al., 2010; Alin et al., 2011; Butman and Raymond, 2011; Dai et al., 2012; Striegl et al., 2012; Abril et al., 2015; Lauerwald et al., 2015; Long et al., 2015). There are no data about $CO_2$ evasion but only $pCO_2$ and average efflux data in Mekong River (Alin et al., 2011; Long et al., 2015), the author estimated $CO_2$ evasion by multiplying the average efflux ($gC/m^2/yr$) and water area ($km^2$). It is important that the model captured a variety of carbon budgets in these six rivers of the globe as depicted by Tranvik et al. (2009). The model result also showed that there was a

great difference of carbon cycle between boreal, temperate, and tropical regions.

In boreal region, the model result showed that about half of annual carbon export occurred in the spring snowmelt period during April – June in Ob and Yukon Rivers (Figs 5a-b) in the same way as Leach et al. (2016). In particular, DOC, POC, and DIC fluxes are higher correlated to surface runoff than groundwater, in particular, in Ob River (correlation between carbon

flux and runoff; $R_{DOC-SW} = 0.42$, $R_{POC-SW} = 0.22$, $R_{DIC-SW} = 0.45$, $R_{DOC-GW} = -0.11$, $R_{POC-GW} = 0.18$, $R_{DIC-GW} = 0.03$, respectively) (Figs 3a-b, and Fig. A3 in Appendix A), and this implies the close relation to biological productivity in the soil-plant system, as suggested by Dinsmore et al. (2013). The simulated results of discharge underestimated the previous data, whereas the simulated horizontal carbon transports generally overestimated the previous materials. The vertical carbon flux of $CO_2$ evasion simulated by the model was in the range of previous data, and became proportionally more important in the early summer

when the downstream carbon export was limited.

In contrast, more carbon is transported during July – September in Yangtze and Mekong Rivers of Southeast Asia due to the effect of monsoon (Figs 5c and e). In Mississippi River, the model result shows more carbon is transported during April – June mainly in the upper Mississippi in the same way as Ob and Yukon Rivers of boreal region, and during July – September mainly

in the lower Mississippi in the same way as Yangtze and Mekong Rivers of subtropical region (Fig. 5d). This is because Mississippi River goes through the wide range of climate zone from the north in boreal to the south in subtropical region (difference of latitude = 20°35'54'' calculated by GTOPO30, HYDRO1K, and ArcGIS v10.1 software). The simulated DOC in Mekong was underestimated in Fig.5e, but this is caused by the inaccuracy of model result and by the limited observed data.




Because there is only one data about Mekong River in the previous material, there is no error bar (range of data) in this figure. Though the simulated DOC in Yangtze and Mississippi was a little underestimated in Figs 5c and d, they were in the range of previous data. The model generally captured the observed data although the simulated DIC and $CO_2$ evasion were underestimated.

In Amazon River, Richey et al. (2002) observed a seasonality in evasion corresponding to both elevated water level and increased $CO_2$ concentration in the river and wetlands. The model showed the higher evasion from large tributaries than from the mainstream channel as shown the author's previous study (Nakayama, 2017a, 2017b), and that the value peaks around the higher water periods during May – June (Fig. 5f). These trends imply the predominance of the seasonality in in-stream

respiration of organic carbon fixed originally on land and along river margins and mobilized into flowing waters, as suggested by the previous observed data (Richey et al., 2002; Rudorff et al., 2011).

### 4.4 Impact of regional hot spots on global carbon cycle change

Fig. 6 shows seasonal and diurnal variations of carbon flux in inland water of six continents simulated by the NICE-BGC. The author has compiled as many previous materials as possible and calculated an averaged value at these regions in Figs 6a-b

(Schlunz and Schneider, 2000; Coynel et al., 2005; Aufdenkampe et al., 2011; Dai et al., 2012; Raymond et al., 2013; Hartman et al., 2014; Borges et al., 2015) in order to estimate $CO_2$ evasion to the atmosphere, sediment storage, and TOC, DOC, POC, DIC-flux to the ocean in previous data. Because the new model can simulate both horizontal transport to the ocean and vertical fluxes and includes aquatic metabolism and terrestrially derived carbon together in major rivers, the result implied that there is active interplay between inorganic and organic carbon (DOC, POC, DIC, $pCO_2$, etc.) through inland waters (Figs 6a-b), as

suggested by the author's previous researches (Nakayama, 2016, 2017a, 2017b). While there was some scattering of the simulated result in comparison with the range of compiled data, such as overestimation of TOC and DOC greatly affected by the values in South America, it is interesting that there was great variability of the seasonal carbon budget among each continent, being greatly affected by various hydro-meteorological conditions, land use, topography, and latitudinal difference, etc.; more organic carbon is exported where wetlands are located in humid tropical and non-carbonate boreal regions (for example, TOC

flux = 299.78 TgC/yr in South America, and = 134.74 TgC/yr in Asia), and more inorganic carbon is exported in areas of high carbonate weathering, soil respiration, and groundwater flow in temperate and boreal regions of carbonate terrain, particularly Africa (DIC flux = 171.85 TgC/yr) (Wang et al., 2013; Borges et al., 2015).

The model result also showed that there is a difference of $CO_2$ evasion and sediment storage between daytime and nighttime

(Figs 6c-d). Though the errors are relatively large, these might be much smaller and we can see statistically significant differences between day and night even for any of the continental depictions if we use the standard error of mean instead of the standard deviation. The author calculated both daytime and nighttime fluxes, which were defined as 8 A.M. to 8 P.M. and as 8 P.M. to 8 A.M. respectively in this study. The result is greatly affected by the fact that $CO_2$ evasion increases and sediment



storage decreases in nighttime (Figs 4d-f), in particular, in Asia and Africa ($CO_2$ evasion increase and sediment storage decrease are 6.35 and 1.32 TgC/yr in Asia, and 5.67 and 0.73 TgC/yr in Africa, respectively). This result also suggests clearly that measurement $pCO_2$ during daytime only, in most of previous observation data, might seriously underestimate annual averages of $CO_2$ outgassing fluxes from inland water to the atmosphere (Peter et al., 2014), and that it is necessary to observed

$pCO_2$ during nighttime furthermore not only in local scale but also in regional scale.

## 5 Discussion

This study implied that seasonal and diurnal variations in biologic processes responsible for production within the terrestrial ecosystem are not negligible on the stream carbon export while runoff plays a dominant control on stream carbon export from the catchment, as pointed out by Leach et al. (2016). There was a clear difference of carbon cycle between Ob River in boreal

region and Mekong River in subtropical region (Fig. 3 and Table 3). The result showed the sensitivity to the resolution of the drainage network used by the model, and big changes to fluxes as a result of the change in spatial scale, which depend on latitude, meteorological conditions, topography, and hydrologic cycle, etc. The model also suggested whether surface water or groundwater is predominant in hydrologic cycle have primarily a great effect on the dominance of organic or inorganic carbons in the basin (Hope et al., 1994). While vast wetland and low topography cause a gradual runoff dominated by the surface water

particularly during snowmelt period in Ob River (Fig. A3 in Appendix A), much rainfall effected by monsoon causes a greater runoff in Mekong River (Fig. A4 in Appendix A). These simulated results also help to hold the first hypothesis described in the section 1 in case of applying the new model in different latitudinal regions. Although the model was verified by using the previous datasets and materials as much as possible (Fig. A2 in Appendix A), the data is rather lacking in the groundwater of global scale except for some data like GLORICH (Hartman et al., 2014), which are a missing point for accurate evaluation of

the carbon budget. It is further necessary for hydrologic and carbon observations both in surface water and groundwater in the future.

While there is some scattering of simulated carbon flux in some large rivers representative of boreal, temperate, and tropical/subtropical regions, the model could reproduce reasonably the previous materials as much as possible (Fig. 5, and Fig.

A2 in Appendix A). In particular, the observation in Amazon River showed that there is a pronounced seasonality in $CO_2$ evasion corresponding to both the elevated water levels and the increased $CO_2$ concentration during Apr.-Jun. though limited sample data for $pCO_2$ and high variations of discharge (Richey et al., 2002), which was reproduced reasonably by NICE-BGC (Fig. 5f). It is further necessary to improve the accuracy of local heterogeneity of carbon flux in the model by coupling or assimilating with ground-truth data, various satellite data to detect greenhouse gases (GHGs) such as GOSAT (Greenhouse

gases Observing SATellite) (Basu et al., 2013) and OCO-2 (NASA's Orbiting Carbon Observatory-2) (Morello, 2014), and isotope analysis to evaluate the age of transported carbon in rivers (Marwick et al., 2015), et al. about the higher resolution of input data such as river network of HydroSHEDS (U.S. Geological Survey, 2013), land cover (Gong et al., 2012), soil,





vegetation, and geology, and the boundary conditions of various chemical variables. Improvement of seasonal carbon flux also depends on a compilation of the seasonal or diurnal fluxes in the previous materials as much as possible (Peter et al., 2014; Leach et al., 2016), a new construction of observation network in spatio-temporal high resolution, and the further verification of the simulated fluxes with them in addition to annual-averaged data.

The model clarified the diurnal dynamics of carbon cycle, in particular, day-night difference in $CO_2$ evasion and the additional change in sediment storage among each river and continent (Figs 4 and 6). The simulated result showed annual-averaged evasion flux during night exceeded that during day by up to 1.17 times at most in Africa (Fig.6c). This value was smaller than the previous measurement in temperate region (Peter et al., 2014), which showed averaged evasion flux during night exceeded

that during day by up to 1.8 times. However, in local scale, the simulated evasion flux during night exceeded that during day by up to 1.43, 3.92, and 1.76 times temporarily in summer in Ob, Yangtze, and Mekong Rivers (Figs 4d-f). This result implies that there are some hot spots and hot moments in the day-night difference of $CO_2$ evasion in rivers. The high ratio might be also related to the impact of human activity on carbon cycle changes in southeast Asia. These results also show that it holds the second hypothesis described in the section 1 in case of applying the new model to regional and global scales. The difference

between the simulated result and the previous study (Peter et al., 2014) is also caused by the reason that their study in the Alpine stream defined daytime as 9 A.M. to 7 P.M. and nighttime as 9 P.M. to 7 A.M., and summer as May – September and winter as October – April, while this paper calculated defined daytime as 8 A.M. to 8 P.M. and nighttime as 8 P.M. to 8 A.M., respectively. Figs 4d-f shows that evasion flux during night might become much larger than that during day temporarily if the author uses the same definition as Peter et al. (2014). It is a future work to use the different day/night definition across a wide

range of latitudes. Anyway, this result sheds light on the further need of implementation of carbon observation network during daytime and nighttime in the future, in particular, in the hot spot of $CO_2$ emission such as Amazon River, central Africa, and south Asia, etc. (Raymond et al., 2013; Lauerwald et al., 2015; Nakayama, 2016). It is also necessary to improve the resolution in the simulation, and to further verify the model results using various field observations, remotely-sensed imagery, and satellite datasets as much as possible in the same way as the author's previous studies about regional hydrologic cycle

(Nakayama and Watanabe, 2004, 2008; Nakayama, 2008). NICE-BGC incorporates the mechanism of $CO_2$ degassing affected by both terrestrially derived $CO_2$ and $CO_2$ production through aquatic metabolism, which were usually evaluated separately in the previous researches except Hotchkiss et al. (2015). The present study has a great improvement from them to support qualitatively the importance of evasion flux during night by developing an advanced model.

Gas evasion from lakes and reservoirs should be accounted in order to evaluate more accurately the total flux from inland water (about $0.32 - 0.64$ PgC/yr) (Aufdenkampe et al., 2011; Raymond et al., 2013). While there are relatively many researches about the estimate of flux from lakes (Alin and Johnson, 2007; Holgerson and Raymond, 2016), there is a large uncertainty about the flux from reservoirs and greatly affected by their age and latitude, etc. (Louis, et al., 2000; Barros et al., 2011). In particular, it is expected that there will be a great change in carbon cycle in Mekong River (Figs 4c and 5e) because many





dams are planned or under construction from now on (Grumbine et al., 2012). It is also possible to clarify more robust partitioning between natural and anthropogenic carbon fluxes in freshwater systems (Ciais et al., 2013) by using this improved model. The high increase of nutrient input such as nitrogen and phosphorus primarily caused by the anthropogenic activity such as fertilizer use and industrial/municipal pollutions (Nakayama, 2015) also change indirectly carbon cycle in Yangtze

River in the future (Figs 4b and 5c). It is further necessary to conduct a sensitivity analysis of timing and volume of irrigation and fertilizer application at 17 major crops in NICE-BGC in order to assess human impacts on the Asian river systems in details in the future. This verification would also help to clarify why and how the more polluted water gives the stronger emission of $CO_2$ in estuaries (Frankignoulle et al., 1998). Because NICE-BGC simulates interplay between inorganic and organic carbon and its relationship to nitrogen and phosphorus, etc. only in fresh water, it is further necessary to clarify the

mechanism of carbon cycle and improve the accuracy of modelled carbon budget in the mouth of the large rivers and estuaries from the view point of the continuum of land-ocean aquatic systems (Regnier et al., 2013) (Fig. 5). In particular, $CO_2$ emission from saline lakes and estuaries also have a significant role in global carbon cycling (Duarte et al., 2008). But there is some research that demonstrates that evasion declines as lakes and freshwater become polluted (Pacheco et al., 2013). This means the future need of model improvement to incorporate ocean model in order to include the effect of seawater intrusion in inland

water. The previous research clarified $pCO_2$ calculated from other parameters under equilibrium condition for a set of input conditions was overestimated in acidic, poorly buffered and organic-rich waters depending on the values of the constant coefficients employed in the model (Abril et al., 2015). Because NICE-BGC also used CO2SYS submodel to calculate $pCO_2$ in the water as mentioned in the section 3, this uncertainty will be further increased in acidic waters, affect directly the calculation of $CO_2$ evasion, and finally attribute to the inaccuracy of carbon cycle in inland water. Future climate change will

have a great effect on the carbon flux in West Siberia through melting of permafrost layer and methane bubble in the peatland (Figs 4a and 5a). These previous researches mean that emissions are spatio-temporally variable or biased between different analyses, and sometimes promoted or sometimes underestimated depending on various conditions such as nutrient content, water depth, DOC, chlorophyll a, dissolved oxygen, etc. These should be incorporated into the model to improve the accuracy of simulated result both in temporal and spatial scales.

**6 Conclusions**

The new process-based model NICE-BGC incorporates the whole process of carbon cycling including surface runoff, groundwater, weathering, $CO_2$ evasion and sediment storage in inland water, and outflow to the ocean. The author evaluates the difference of hydrologic and carbon cycles between Ob River, Yangtze River, and Mekong River basin in Asia by using different resolutions of river network data. The model simulated more heterogenous distributions of water and carbon flux in

the finer river network data in these regions, and clarified the latitudinal effect and human impact on carbon cycle change and helped to identify some hot spots on a regional scale. Then, the model was extended to the regional and continental scales at 1°x1° resolution with a time step of $\Delta t = 1$ day to evaluate seasonal and diurnal variations of carbon cycle in terrestrial-aquatic





continuum. The model result showed that there is a great variability of horizontal transport to the ocean and vertical fluxes among boreal, temperate, and tropical regions and among each continent, which reflects diurnal and seasonal variations of biologic and hydrologic processes there. In particular, the result of diurnal cycle showed $CO_2$ evasion increases and sediment storage decreases in nighttime, particularly clearly seen temporarily in summer in Yangtze River, which implied that there are

some hot spots and hot moments in the day-night difference of vertical fluxes in regional scale. These results also emphasize the important role of Asian river systems on global carbon cycle and the further need to improve the resolution of the simulation, to implement carbon observation network, and to apply satellite data in the higher-resolution in the future.

**Acknowledgements**

Data management support for preparing, documenting, and model simulation, and output data was performed by the Center

for Global Environment Research (CGER) at the National Institute for Environmental Studies (NIES), Japan. All the data used in this paper are available upon request to the corresponding author (E-mail address: nakat@nies.go.jp). Dr. S. Maksyutov, National Institute for Environmental Studies (NIES), gave the author helpful discussions about the global carbon cycle. The author thanks Dr. R. Wania, Austria, for advice about LPJWHyMe, Dr. G. Pelletier, Dept. of Ecology, USA, for advice about QUAL2K, Dr. R. Srinivasan, Texas A&M University, USA, for advice about SWAT, and Dr. A. Ridgwell, University of

Exeter, UK, for advice about RokGeM. These were of great help for development of the NICE-BGC. The author also thanks Dr. J. Hartmann, Universität Hamburg, Germany, for providing the Global River Chemistry Database (GloRiCh). Dr. W. Ludwig, Universite de Perpignan, France, and Dr. P. Amiotte-Suchet, Universite de Bourgogne, France, provided the ISLSCP II data.

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





| Parameter in QUAL2Kw | Unit in QUAL2Kw | Input from Hillslope | Input from Groundwater | Unit in Input Parameter |
|---|---|---|---|---|
| Inflow | m3/s | SURQ | GW_Q | mm |
| Temperature | C° | TMP | 15.0 | C° |
| Conductivity | µS/cm 25C° | 100.0 | 100.0 | µS/cm 25C° |
| Inorganic Suspended Solids | mgD/L | SYLD | 0.0 | metric tons/ha |
| Dissolved Oxygen (DO) | mgO2/L | DOXQ | DOXQ | kg O2 |
| Slow CBOD | mgO2/L | CBODU/2 | 0.0 | kg/cbod |
| Fast CBOD | mgO2/L | CBODU/2 | 0.0 | kg/cbod |
| Organic Nitrogen | µgN/L | ORGN | 0.0 | kg N/ha |
| Ammonia Nitrogen (NH4+) | µgN/L | 0.0 | 0.0 | kg N/ha |
| Nitrate Nitrogen (NO3-) | µgN/L | NSURQ | LATQ + GWNO3 | kg N/ha |
| Organic Phosphorus | µgP/L | ORGP | 0.0 | kg P/ha |
| Inorganic Phosphorus | µgP/L | SOLP + SEDP | GWSOLP | kg P/ha |
| Phytoplankton | µgA/L | POC | 0.0 | mg/L |
| Detritus | mgD/L | POC | 0.0 | mg/L |
| Pathogen | cfu/100 mL | 0.0 | 0.0 | cfu/100 mL |
| Alkalinity | mgCaCO3/L | ALK | ALK | µmol/kg |
| pH | - | PH | PH | - |

**Table 1: List of input parameters from terrestrial ecosystems into the revised QUAL2Kw.**

| Stream Order | West Siberia | | | Mekong Riv. | | |
|---|---|---|---|---|---|---|
| | HYDRO (12 reaches) | HYDRO (128 reaches) | GRDC (1160 reaches) | HYDRO (19 reaches) | HYDRO (82 reaches) | HYDRO (449 reaches) |
| 5 | 0 | 14 | 0 | 0 | 12 | 226 |
| 6 | 1 | 33 | 647 | 0 | 27 | 100 |
| 7 | 2 | 32 | 270 | 10 | 21 | 86 |
| 8 | 3 | 22 | 164 | 5 | 7 | 17 |
| 9 | 3 | 12 | 55 | 4 | 15 | 20 |
| 10 | 3 | 15 | 24 | 0 | 0 | 0 |

**Table 2: Number of segments in stream order of each network data for the simulation shown in Figure 2. Stream order is according to Downing et al. (2012).**





| Case | Area | SW-Inflow (km3/yr) | GW-Inflow (km3/yr) | Total-Inflow (km3/yr) | DOC-flux (TgC/yr) | POC-flux (TgC/yr) | DIC-flux (TgC/yr) | Total-flux (TgC/yr) |
|---|---|---|---|---|---|---|---|---|
| HYDRO (12 reaches) | West Siberia | 175.0 | 51.745 | 226.761 | 17.527 | 1.972 | 12.259 | 31.757 |
| HYDRO (128 reaches) | West Siberia | 228.1 | 69.003 | 297.112 | 23.913 | 1.884 | 15.882 | 41.679 |
| GRDC (1160 reaches) | West Siberia | 239.7 | 76.641 | 316.381 | 33.797 | 2.464 | 16.663 | 52.925 |
| HYDRO (19 reaches) | Mekong Riv. | 94.0 | 668.159 | 762.204 | 1.926 | 1.557 | 19.970 | 23.452 |
| HYDRO (82 reaches) | Mekong Riv. | 78.5 | 619.794 | 698.295 | 1.571 | 9.093 | 16.139 | 26.803 |
| HYDRO (449 reaches) | Mekong Riv. | 77.0 | 623.412 | 700.389 | 2.259 | 4.892 | 13.738 | 20.889 |

**Table 3: Comparison of annual runoff and carbon flux in three types of river data in Ob River and Mekong River basins shown in Figure 2.**






**Figure 1: Geographical characteristics and elevation in the study area of Ob, Yangtze, and Mekong River basins representative of Asian river systems.**




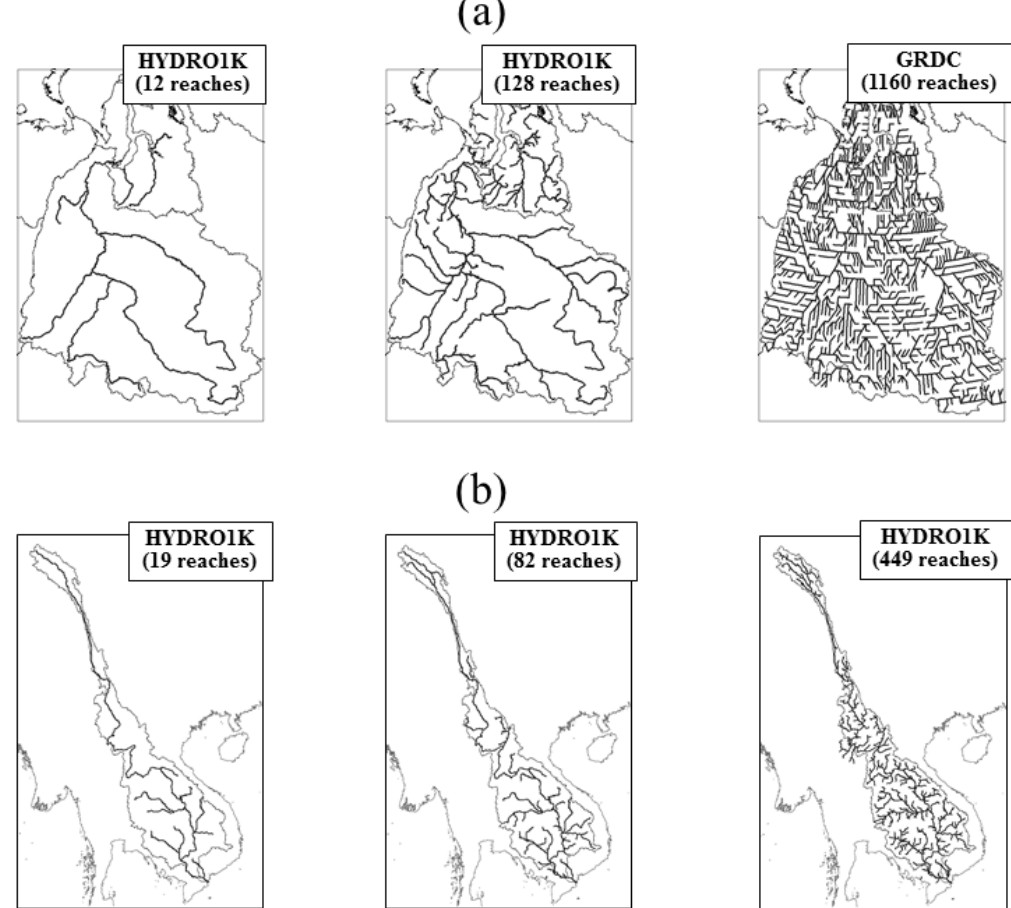

**Figure 2: Comparison of network data for the simulation; (a) West Siberian Lowland (12 , 128, and 1160 stream channels), and (b) Mekong River basin (19, 82, and 449 stream channels).**





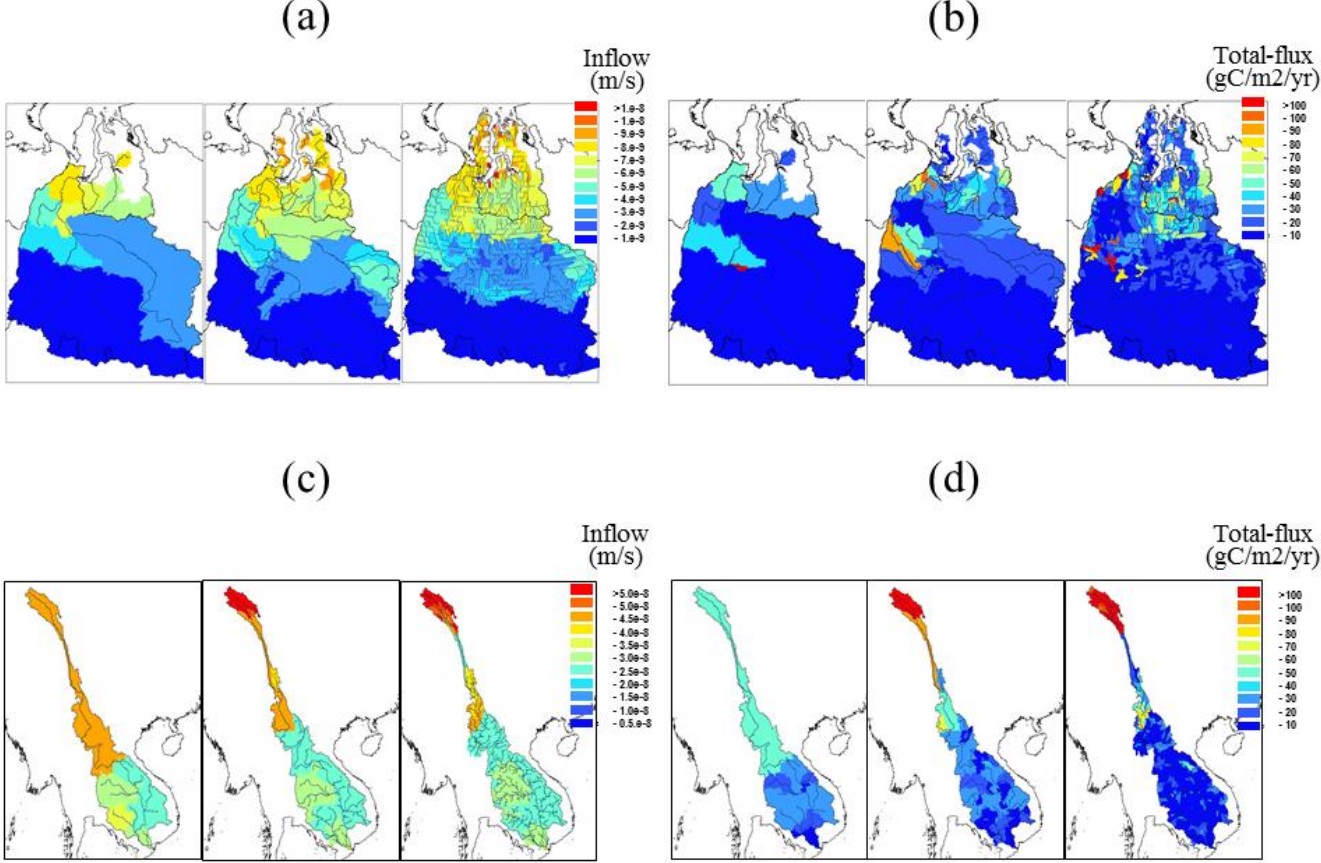

**Figure 3: Comparison of annual-averaged water and carbon flux from terrestrial into aquatic ecosystems in three types of river data (Figure 2) simulated by NICE-BGC in West Siberia; (a) total inflow (= surface water + groundwater), (b) TOC flux (= DOC + POC + DIC flux), and in Mekong River; (c) total inflow, (d) total flux, respectively.**





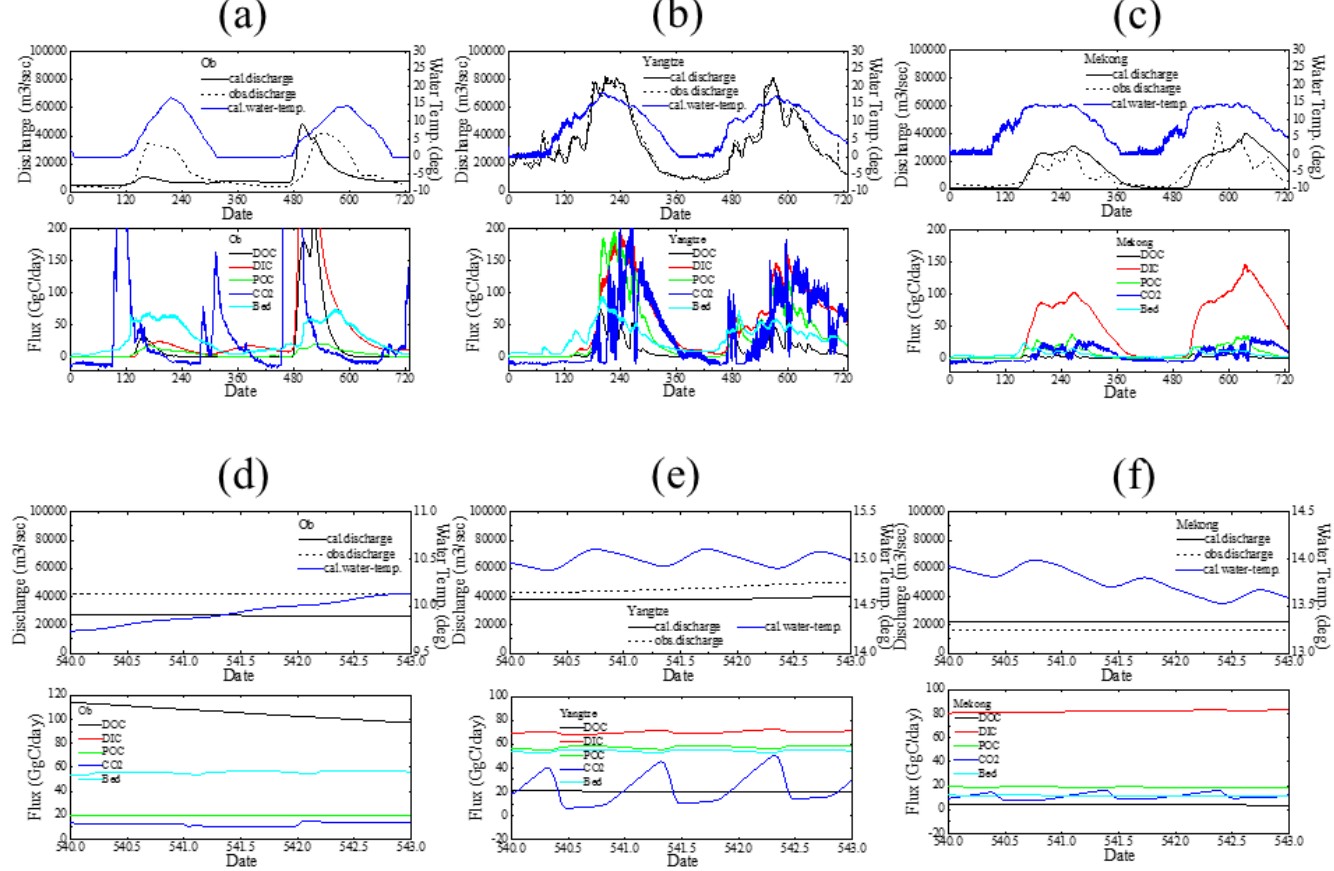

**Figure 4: Simulated result of time-series of discharge and carbon flux in Ob (boreal), Yangtze (temperate), and Mekong (tropical/subtropical) Rivers representative of Asian river systems in; (a)-(c) two years during 1998 – 1999, and (d)-(f) three days during June 24 – 27 in 1999. In the above one of Figures 4a-f, solid-line is the simulated discharge, dashed-line is the observed discharge (Changjiang Water Conservancy Committee, 1998, 1999; Lammers and Shiklomanov, 2000; Mekong River Commission, 2011), and bold-line is the simulated water temperature. In the upper one of Figures 4a-f, black line, bashed line, and blue line mean calculated discharge, observed discharge, and calculated water temperature, respectively. In the below one of Figures 4a-f, black line, red line, green line, blue line, and light-blue line mean DOC, DIC, POC, CO₂ evasion, and sediment storage, respectively.**



**Figure 5: Seasonal variations of carbon flux in six major rivers around 2000 simulated by NICE-BGC; boreal in (a) Ob and (b) Yukon, temperate in (c) Yangtze and (d) Mississippi, and tropical/subtropical in (e) Mekong and (f) Amazon Rivers, respectively. "TOC", "DOC", "DIC", "POC" mean carbon flux outflow to the ocean, "CO2" means CO₂ evasion from inland water to the atmosphere, "Bed" means sediment storage, and "Q" means discharge to the ocean in each river. Circle shows a mean value compiled by previous materials, and error bar shows the range of data.**





**Figure 6: Carbon flux budgets in inland water around 2000 simulated by NICE-BGC; (a) seasonal change in total river discharge to the ocean in each continent, (b) global carbon budget in each continent, (c) day-night difference in CO₂ evasion of each continent, and (d) day-night difference in sediment storage of each continent, respectively. "AS", "EU", "OC", "AF", "NA", "SA" mean Asia, Europe, Oceania, Africa, North America, and South America, respectively. In Figures 6a-d, error bar in bold dotted-line shows the standard deviation of annual-averaged values simulated by the model. In Figures 6a-b, circle shows a mean value compiled by previous materials, and error bar shows the range of data. In Figures 6c-d, "-D" and "-N" mean daytime and nighttime values, which were defined as 8 A.M. to 8 P.M. and as 8 P.M. to 8 A.M., respectively.**



**Appendix A**

This Appendix provides supporting figures. Figure A1 shows development of eco-hydrologic and biogeochemical coupling model along terrestrial-aquatic continuum (NICE-BGC) in the author's previous paper (Nakayama, 2017a). Figure A2 shows the model verification of water and carbon flux in 27 different rivers. Figures A3 and A4 show annual-averaged water and carbon flux from terrestrial into aquatic ecosystems in three types of river data simulated by the model in West Siberia and Mekong River, and implies the interplay between surface water and groundwater, and organic and inorganic carbons, respectively.

**Figure A1: Flow diagram of new development of eco-hydrologic and biogeochemical coupling model along terrestrial-aquatic continuum (NICE-BGC) (Nakayama, 2017a).**





**Figure A2: Comparison of (a) river discharge, (b) CO₂ evasion in inland water, and annual carbon flux to the ocean between NICE-BGC and previous data in 27 different rivers and some regions; (c) TOC, (d) DOC, (e) POC, and (f) DIC (Nakayama, 2017a). Error bar shows the range of data compiled by previous materials. Dotted line means NICE-BGC value is equal to the reference value.**



**Figure A3: Comparison of annual-averaged water and carbon flux from terrestrial into aquatic ecosystems in three types of river data in West Siberia; (a) surface water, (b) groundwater, (c) DOC flux, (d) POC flux, and (e) DIC flux.**





**Figure A4: Comparison of annual-averaged water and carbon flux from terrestrial into aquatic ecosystems in three types of river data in Mekong River; (a) surface water, (b) groundwater, (c) DOC flux, (d) POC flux, and (e) DIC flux.**