# Peer review of "Biogeochemical contrast between different latitudes and the effect of human activity on spatio-temporal carbon cycle change in Asian river systems"

_Biogeosciences, 2017_

## Referee Comment (RC1) · Anonymous Referee #1 · 28 Nov 2017

The paper compares the fluxes of carbon (C) in the Ob, Yangtze, and Mekong Rivers, that were found to be very different due to differences in climate (latitude). This conclusion makes sense but was highly predictable since the fluxes of C from major World rivers has been known with certainty for several decades, going back to the papers of Meybeck in the 1980's, that were followed by an abundant literature.

Consequently, the paper is of marginal interest and does not add anything significantly new to the recent papers from the author (who is prolific), such as his two papers published in 2017 (JGR & Ecol. Model.) and 2016 (Ecohydrology and Hydrobiology)

[Figure]

that dealt with C fluxes from World Rivers.

For instance, Figure A2 of the present ms (BGD) is exactly identical to Figure 3 of the JGR 2017 paper, and nearly identical (except one panel) to Figure 6 of the Ecohydrology and Hydrobiology 2016 paper. Figure 4 of the 2017 JGR paper is nearly identical to Figure 6 of the 2017 Ecol. Model. paper, that have been re-arranged slightly to produce Figure 6 of the present BGD ms (only a minor cosmetic re-arrangement of the figure, the actual content of the plots is identical).

More importantly, the paper as it stands does not actually address the topic of the special issue that deals with "Human impacts on carbon fluxes in Asian river systems". However, the author has a modelling tool (NICE-BGC) that could be very useful to address this topic.

First, the author could compare a reference simulation (without dams) for all of the major Asian Rivers with a simulation including water diversion by dams. This would allow to estimate how dams have already affected the C fluxes in Asian Rivers (some of which are the largest in the World, such as the Three Gorges).

Second, in a similar fashion, the authors could estimate the future impact on C fluxes in Asian Rivers from the building of planned infrastructures. For instance, enormous dams are planned for construction in the Mekong River network that are expected to severely affect suspended matter load with potential problems in the Mekong delta (erosion).

Construction of dams will not only affect TSM and C fluxes but also the delivery of nutrients to the East China Sea, with potential impacts on its productivity.

Third, the author could use climate predictions for the next 100 yrs in the region and estimate how predicted changes in precipitation and temperature could affect the C and nutrient fluxes from Asian Rivers. This would allow to address the question whether building of dams or climate change would most affect the C fluxes in Asian Rivers.
Addressing the above three points would be spot on the topic of the Special Issue, and would be a useful and interesting application of the NICE-BGC model.

If the author decides to modify the paper in this direction, then all of the parts of the paper regarding validation at global scale (comparison with Amazon, Yukon, etc . . .) should be removed. On the other hand, the author should focus on a careful validation of the Asian rivers, for which a large amount of data is available from literature. For instance, I would like to see how the NICE-BGC model can simulate seasonal variations of concentrations of DOC, POC, TSM, HCO3-, CO2, etc compared to seasonal changes of these quantities from observations in the different Asian Rivers (available from literature).

As it stands the validation of the model given in Fig. 5 is somewhat worrying. The modelled DOC flux from the Mekong is under-estimated by a factor of 10 compared to field data, while the modelled DIC flux is over-estimated by nearly a factor of 4. For Yangtze, similar large discrepancies are also observed, for instance the modelled POC flux is over-estimated by a factor of 3.

---

## Author Comment (AC1) · 21 Dec 2017

The author thanks to the referee's valuable comments and revised the manuscript according to them in the following. These revisions also help to clarify its effect of human activity on spatio-temporal carbon cycle change in Asian river systems, which would be spot on the topic of the Special Issue and would be a useful and interesting application of the NICE-BGC model. Detailed response is described in the following; ——————————————-

[Figure]

- The paper compares the fluxes of carbon (C) in the Ob, Yangtze, and Mekong Rivers, that were found to be very different due to differences in climate (latitude). This conclusion makes sense but was highly predictable since the fluxes of C from major World rivers has been known with certainty for several decades, going back to the papers of Meybeck in the 1980's, that were followed by an abundant literature. Consequently, the paper is of marginal interest and does not add anything significantly new to the recent papers from the author (who is prolific), such as his two papers published in 2017 (JGR & Ecol. Model.) and 2016 (Ecohydrology and Hydrobiology) that dealt with C fluxes from World Rivers. For instance, Figure A2 of the present ms (BGD) is exactly identical to Figure 3 of the JGR 2017 paper, and nearly identical (except one panel) to Figure 6 of the Ecohydrology and Hydrobiology 2016 paper. Figure 4 of the 2017 JGR paper is nearly identical to Figure 6 of the 2017 Ecol. Model. paper, that have been re-arranged slightly to produce Figure 6 of the present BGD ms (only a minor cosmetic re-arrangement of the figure, the actual content of the plots is identical).

Ans. The author generally agrees to the referee's comments. Yes, he used Figure A1 and A2 in Appendix in order to show more clearly the general structure of the model and the model validation in global rivers in the previous papers. So, he rewrote as the reprint from the previous papers in the figure captions after copyright permission.

————————————————-

- More importantly, the paper as it stands does not actually address the topic of the special issue that deals with "Human impacts on carbon fluxes in Asian river systems". However, the author has a modelling tool (NICE-BGC) that could be very useful to address this topic. First, the author could compare a reference simulation (without dams) for all of the major Asian Rivers with a simulation including water diversion by dams. This would allow to estimate how dams have already affected the C fluxes in Asian Rivers (some of which are the largest in the World, such as the Three Gorges). Second, in a similar fashion, the authors could estimate the future impact on C fluxes in Asian Rivers from the building of planned infrastructures. For instance, enormous

dams are planned for construction in the Mekong River network that are expected to severely affect suspended matter load with potential problems in the Mekong delta (erosion). Construction of dams will not only affect TSM and C fluxes but also the delivery of nutrients to the East China Sea, with potential impacts on its productivity. Third, the author could use climate predictions for the next 100 yrs in the region and estimate how predicted changes in precipitation and temperature could affect the C and nutrient fluxes from Asian Rivers. This would allow to address the question whether building of dams or climate change would most affect the C fluxes in Asian Rivers. Addressing the above three points would be spot on the topic of the Special Issue, and would be a useful and interesting application of the NICE-BGC model.

If the author decides to modify the paper in this direction, then all of the parts of the paper regarding validation at global scale (comparison with Amazon, Yukon, etc : : :) should be removed. On the other hand, the author should focus on a careful validation of the Asian rivers, for which a large amount of data is available from literature. For instance, I would like to see how the NICE-BGC model can simulate seasonal variations of concentrations of DOC, POC, TSM, $HCO_3^-$, $CO_2$, etc compared to seasonal changes of these quantities from observations in the different Asian Rivers (available from literature).

Ans. The author thanks very much to the referee's valuable comments. He generally agrees three points suggested by the referee are important for the topic of the Special Issue. However, the time is very limited for the revision of the paper (about one month) according to these three points. The author has already submitted another paper and now under review about the first point in global scale. The second point is also beneficial, and the author is now preparing to do this in the future work. Third point is very important, but actually it takes more time to do a climate prediction for the next 100 years. Of course, he agrees well with the importance of addressing the question whether building of dams or climate change would most affect the carbon fluxes in Asian Rivers. But it is rather a future work to carry out a comprehensive analysis based on these three points including the effect of land use change. Instead, it is necessary to put together the previous literature as much as possible, and clarify the actual state of carbon cycle in Asian Rivers and this effect on global carbon cycle. So, he did not delete Figure 6 because he thinks this figure is important to estimate them. Otherwise, he worries that the paper would become an incoherent structure including all three points before the clarification of the actual state of carbon cycle in Asian Rivers. For the above reasons, the author removed the validation at global scale in Figure 5 according to the referee's comment. Then, he extended the simulation period for 18 years in order to validate the simulated result with the limited previous literature as much as possible. He gathered and added more literature about time-series and seasonal variations of DOC fluxes as much as possible to compare seasonal changes of these quantities in different Asian Rivers in the revised Figure 4. In particular, he compared the available literature with the simulated result in Ob River (Kohler et al., 2003; Gebhadrt et al., 2004; McClelland et al., 2016), Yangtze River (Wang et al., 2012; Shi et al., 2016; Ran et al., 2017), and Mekong River (Li and Bush, 2015; Borges et al., 2017) as much as possible though there were not so many available data except DOC yet. This validation also improved the accuracy of carbon flux in the revised Figure 5, as described in the following comment. The author also revised the manuscript to explain the above in details. ————————————————-

As it stands the validation of the model given in Fig. 5 is somewhat worrying. The modelled DOC flux from the Mekong is under-estimated by a factor of 10 compared to field data, while the modelled DIC flux is over-estimated by nearly a factor of 4. For Yangtze, similar large discrepancies are also observed, for instance the modelled POC flux is over-estimated by a factor of 3.

Ans. The author added newly Table 4 about some literature further to constrain carbon budget in Ob River (Gordeev and Kravchishina, 2009; Mann et al., 2016; McClelland et al., 2016; Kaiser et al., 2017), Yangtze River (Li et al., 2012; Shi et al., 2016; Guo et al., 2015; Gao et al., 2017; Ran et al., 2017), and Mekong River (Li and Bush, 2015;

Borges et al., 2017) in the revised Figure 5, respectively. Because the author added to validate the time-series and seasonal variations as described in Figure 4, the model result improved the accuracy in Figure 5. ——————————————-

Please also note the supplement to this comment:
https://www.biogeosciences-discuss.net/bg-2017-447/bg-2017-447-AC1-supplement.zip

---

## Referee Comment (RC2) · Anonymous Referee #2 · 5 Jan 2018

1.. The objectives of this manuscript are to evaluate 2 hypotheses for the Ob, Yangtze, and Mekong Rivers: (i) Latitudinal difference is an important factor to constrain a variability of horizontal and vertical carbon fluxes, and (ii) Seasonal and diurnal variations are important components to evaluate the impact of regional hot spots on global carbon cycle change. The toolset is to apply the NICE-BGC model, developed by the author.

2. I have two overall levels of response/review. Level #1. As a pure modeling exercise. Level #2. As contributing knowledge to the biogeochemistry of river carbon cycles in these 3 rivers, and how this knowledge might be extrapolated; i.e, addressing the

hypotheses.

3. Level #1. As a modeling exercise. The NICE-BGC model is very complex, addressing pretty much the entire land-surface dynamics of ecosystem function, and water and carbon dynamics. This brings up multiple complexities in data acquisition and processing (that I am painfully aware of), accurately expressing the processes being addressed, and then the sheer computational logistics. Very challenging – I applaud the author for even taking this on.

Several points: (1) The devil is in the details, especially with regard to how specific processes are represented in the model. Simply that there is an algorithm with the name eg "CO2 evasion" doesn't tell us anything – what is it? This becomes important at multiple levels – eg how is sediment storage or outgassing represented with enough resolution to distinguish between day and night fluxes?

(2) An important issue is to make sure all of the scales and time steps line up for the processes represented. Yes, he did dial through the several different resolutions for Hydro, which was interesting. But more broadly there is often disconnect between scale of forcings, landscape, and time step of process. A 1o x 1o scale is ~100km a side, which implies a travel time for water quite in excess of diurnal processes.

(3) Calibration/validation is, indeed, a significant challenge. A thoughtful evaluation of the problem would be useful. I certainly appreciate that data can be sparse, especially to represent all of the processes included in the model. But there is much more available than is used here, which can lead to erroneous conclusions (some, below)

4. Level #2. Biogeochemistry

I find the conclusions on the biogeochemistry to be either not at all surprising, and often disagree.

(1) The first step is to units straight. Fluxes are often referred to gC m-2 yr-1. Per m-2 of what? The total area of the basin (eg Stung Treng, for Mekong) or per local tributary,

or per ??? Without knowing this is impossible to evaluate. (2) The next step is for the reader to have some confidence in computed results. A thorough cal/val process would be useful here, but the manuscript doesn't do that. So do presented results make at least intuitive sense? Table 3 says that GW is ~89% of total flow of the Mekong. I would be shocked if that is true (I've co-authored papers on Mekong hydrology) - it has to be wrong. Results for the Amazon had discharge had perhaps a low-water average, not annual, while TOC fluxes were far too low. Mekong water temperatures are far too cold (why do T)? Etc etc. (3) I don't understand why trying to do diurnal fluxes, Yes, there can be differences in pCO2 concentrations in still waters, much less so in rivers, usually mitigated by lower wind speeds, but this isn't relevant at more regional scales, and don't understand how the model can represent it. I have no idea how to get/explain diurnal variation in sediments (outside of tides, but that is a different issue). If real, I would be most interested in seeing that documented. (4) Why do the 6 other rivers? Not part of the original argument. (5) Etc

5. Evaluation: This manuscript, while potentially very interesting, is another example of the difficulties in taking complex more global models, and using them to interpret finer scale dynamics. Such models certainly have their place, but cuation must be taken in their application.

(1) The manuscript doesn't address at all the stated objective of human activity (2) Yes, there are differences between the 3 main rivers over time and space, but the manuscript doesn't attribute cause/effect with any confidence. (3) The toolset used, NICE-BGC, wasn't applied with enough specificity and confidence, relative to the questions asked. Topics were all over the place, without enough detail to be convincing. It is not clear that the author really understands the details of the biogeochemistry being addressed. To be fair, a very wide range of issues were included, but a better strategy might be to parse those out into several more focused manuscripts.

---

## Author Comment (AC2) · 16 Jan 2018

Detailed Response to Referee #2 Ms. Ref. No.: bg-2017-447 Journal: Biogeosciences Title: Biogeochemical contrast between different latitudes and the effect of human activity on spatio-temporal carbon cycle change in Asian river systems

The author thanks to the referee's valuable comments and revised the manuscript according to them in the following. These revisions also help to clarify the important role of Asian river systems on global carbon cycle, and implies a vulnerability of Asian rivers

to anthropogenic activity and climate change. The results would be spot on the topic of the Special Issue and would be a useful and interesting application of the NICE-BGC model. Detailed response is described in the following;

1.. The objectives of this manuscript are to evaluate 2 hypotheses for the Ob, Yangtze, and Mekong Rivers: (i) Latitudinal difference is an important factor to constrain a variability of horizontal and vertical carbon fluxes, and (ii) Seasonal and diurnal variations are important components to evaluate the impact of regional hot spots on global carbon cycle change. The toolset is to apply the NICE-BGC model, developed by the author. 2. I have two overall levels of response/review. Level #1. As a pure modeling exercise. Level #2. As contributing knowledge to the biogeochemistry of river carbon cycles in these 3 rivers, and how this knowledge might be extrapolated; i.e, addressing the hypotheses.

Ans. The author generally agrees to the referee's comments about two overall levels. He revised the manuscript as much as possible to improve the accuracy of simulated results. In particular, he added some explanations about how this knowledge might be extrapolated by addressing two hypotheses. The quantification of first hypothesis is important in Asian Rivers because riverine carbon fluxes vary between different latitudes, as clearly shown in the new Figure 4. The quantification of seasonal and diurnal variations is also valuable to implement carbon observation network in the future. Detailed responses of these two levels are described in the following.

3. Level #1. As a modeling exercise. The NICE-BGC model is very complex, addressing pretty much the entire land-surface dynamics of ecosystem function, and water and carbon dynamics. This brings up multiple complexities in data acquisition and processing (that I am painfully aware of), accurately expressing the processes being addressed, and then the sheer computational logistics. Very challenging – I applaud the author for even taking this on. Several points: (1) The devil is in the details, especially with regard to how specific processes are represented in the model. Simply that there is an algorithm with the name eg "CO2 evasion" doesn't tell us anything –
what is it? This becomes important at multiple levels – eg how is sediment storage or outgassing represented with enough resolution to distinguish between day and night fluxes? (2) An important issue is to make sure all of the scales and time steps line up for the processes represented. Yes, he did dial through the several different resolutions for Hydro, which was interesting. But more broadly there is often disconnect between scale of forcings, landscape, and time step of process. A 1o x 1o scale is -100km a side, which implies a travel time for water quite in excess of diurnal processes. (3) Calibration/validation is, indeed, a significant challenge. A thoughtful evaluation of the problem would be useful. I certainly appreciate that data can be sparse, especially to represent all of the processes included in the model. But there is much more available than is used here, which can lead to erroneous conclusions (some, below)

Ans. The author thanks very much to the referee's valuable comments. (1) The author thanks to the referee's important comments, and agrees with the need to represent specific processes and algorithms in details. He added some descriptions, in particular, about CO2 evasion and sediment storage, in order to be easily understandable according to the comment. As NICE-BGC simulates CO2 evasion and sediment storage with more smaller time step between ïĄĎt = 0.044 min and 0.70 min for the stability of the model, it is possible to simulate the diurnal water and carbon cycles in inland waters. (2) The model simulates river network among elements and terrestrial ecosystem in grid cells. Then, the contribution of river on grid cell was estimated by calculating area ratio of river (multiplication of river width and length) in each grid. Then, NICE-BGC simulation for aquatic ecosystem was conducted with more smaller time step between ïĄĎt = 0.044 min and 0.70 min for the stability of the model, which made it possible to simulate the diurnal water and carbon cycles in inland waters. So, the time step of the model was less than the travel time for water quality of diurnal processes. Details are described in the section 3.4 of the original manuscript, and the author added some explanation in the revised manuscript. (3) The author added more calibration/validation process to extend the simulation period for 18 years in order to validate the simulated result with the limited previous literature as much as possible (in new Table 4) according to the comments of Referees #1 and #2. This validation also improved the accuracy of carbon flux in the revised Figures 4 and 5.

The author added the above (1)-(3) explanations in the revised manuscript.

4. Level #2. Biogeochemistry I find the conclusions on the biogeochemistry to be either not at all surprising, and often disagree. (1) The first step is to units straight. Fluxes are often referred to gC m-2 yr-1. Per m-2 of what? The total area of the basin (eg Stung Treng, for Mekong) or per local tributary, or per ??? Without knowing this is impossible to evaluate. (2) The next step is for the reader to have some confidence in computed results. A thorough cal/val process would be useful here, but the manuscript doesn't do that. So do presented results make at least intuitive sense? Table 3 says that GW is -89% of total flow of the Mekong. I would be shocked if that is true (I've co-authored papers on Mekong hydrology) - it has to be wrong. Results for the Amazon had discharge had perhaps a low-water average, not annual, while TOC fluxes were far too low. Mekong water temperatures are far too cold (why do T)? Etc etc. (3) I don't understand why trying to do diurnal fluxes, Yes, there can be differences in pCO2 concentrations in still waters, much less so in rivers, usually mitigated by lower wind speeds, but this isn't relevant at more regional scales, and don't understand how the model can represent it. I have no idea how to get/explain diurnal variation in sediments (outside of tides, but that is a different issue). If real, I would be most interested in seeing that documented. (4) Why do the 6 other rivers? Not part of the original argument. (5) Etc

Ans. The author thanks to the referee's valuable comments. (1) The author added an explanation that fluxes are per each basin according to the referee's comment. (2) The author thanks to the referee's important comment about some confidence in computed results. He added more calibration/validation process to extend the simulation period for 18 years in order to validate the simulated result with the limited previous literature as much as possible by adding new Table 4 according to the comments of Referees #1 and #2. This validation also improved the accuracy of carbon flux in the revised

Figures 4 and 5. The author generally agrees with the referee's comment that the ratio of GW is higher in Mekong River basin. However, the greater value for groundwater runoff, compared to surface runoff, does not surprise me, and is what the author would expect. Except in the upper Mekong in the Yunnan Province of China (Himalayas), where it cuts deep gorges into the rock and most of the inflows are probably by surface pathways, the author thinks in the lower Mekong most inflow to the tributary rivers is likely via subsurface. This higher ratio of GW can be seen in the spatial distribution of SW and GW runoff in Figures A4(a)-(c). This higher GW also agrees with the previous model study of Mekong basin upwards from Kratie only that the overflow part of the total flow for the period 1985-2003 was about 18% (Lauri et al., 2012). It should be noted that this value depends on the calibration to adjust some parameters about soil water conductivity on slopes and river friction coefficients, etc. Also human impact of collecting the water in many small reservoirs may affect the overflow percentage. This phenomenon is also related to the fact that there is much more rainfall in the rainy season (June to September) in the lower gradient, agricultural, and vast region of middle-lower Mekong than in the narrow area of upper Mekong (Costa-Cabral et al., 2008). Anyway, subsurface runoff is generally higher than surface runoff in continental or global scale because of the greater reserve of water in groundwater as pointed out by the previous studies (Oki and Kanae, 2006; Good et al., 2015; Nakayama, 2017a). The author also thanks to the referee's comment that he mistook the plot of water temperature in the original figure. Yes, the water temperature in Mekong River is in the range of 20-30 C° and that in Yangtze is in the range of 10-25 C°. These values are reasonable. (3) The author can understand somewhat to say an nonessentiality to try to evaluate diurnal fluxes. However, the result in this paper suggests that measurement $pCO_2$ during daytime only, in most of previous observation data, might seriously underestimate annual averages of $CO_2$ outgassing fluxes from inland water to the atmosphere (Peter et al., 2014), and that it is necessary to observed $pCO_2$ during nighttime furthermore not only in local scale but also in regional scale, as described in the original paper. While this phenomenon is apparent in temperate region of Austria (Peter et al., 2014), it is not

clear whether this is relevant at more regional scale or not because it might depend on various conditions such as climate, hydrology, geology, vegetation, etc. But it is certain that the difficulty to observe/estimate diurnal fluxes might increase as it widens the target area more because of the difficulty to do these fluxes except satellite analysis and modelling approach. From this viewpoint, this author thinks it is valuable to evaluate diurnal fluxes in different scales in order to implement carbon observation network in the future. The same might be true to get/explain diurnal variations in sediments, as can be seen in Figure 6. (4) The author removed the validation of global other rivers and concentrated on three Asian Rivers according to the comment of Referees #1 and #2. (5) The author added some other explanations about biogeochemical processes according to the referee's comments.

The author added the above (1)-(5) explanations in the revised manuscript.

5. Evaluation: This manuscript, while potentially very interesting, is another example of the difficulties in taking complex more global models, and using them to interpret finer scale dynamics. Such models certainly have their place, but cuation must be taken in their application. (1) The manuscript doesn't address at all the stated objective of human activity. (2) Yes, there are differences between the 3 main rivers over time and space, but the manuscript doesn't attribute cause/effect with any confidence. (3) The toolset used, NICE-BGC, wasn't applied with enough specificity and confidence, relative to the questions asked. Topics were all over the place, without enough detail to be convincing. It is not clear that the author really understands the details of the biogeochemistry being addressed. To be fair, a very wide range of issues were included, but a better strategy might be to parse those out into several more focused manuscripts.

Ans. The author also thanks to the referee's important comments. (1) The author can understand the importance of the referee's comment. The author has already submitted another paper and now under review about the impact of human activity (a reference simulation without dams and a simulation including dams) in global scale, as described in the comment of Referee #1. It is rather a future work to carry out a

comprehensive analysis based on this point including the effect of land use change. Instead, it is necessary to put together the previous literature as much as possible, and clarify the actual state of carbon cycle in Asian Rivers and this effect on global carbon cycle. Otherwise, he worries that the paper would become an incoherent structure before the clarification of the actual state of carbon cycle in Asian Rivers. (2) The author generally agrees with the referee's comment and added some quantitative explanations about cause/effect based on the simulated results in the revised manuscript. He extended the simulation period for 18 years in order to validate the simulated result with the limited previous literature as much as possible. He gathered and added more literature about time-series and seasonal variations of DOC fluxes as much as possible to compare seasonal changes of these quantities in different Asian Rivers in the new Table 4 and the revised Figure 4. In particular, he compared the available literature with the simulated result in Ob River (Kohler et al., 2003; Gebhadrt et al., 2004; McClelland et al., 2016), Yangtze River (Wang et al., 2012; Shi et al., 2016; Ran et al., 2017), and Mekong River (Li and Bush, 2015; Borges et al., 2017) as much as possible though there were not so many available data except DOC yet. This validation also improved the accuracy of carbon flux in the revised Figure 5, as described in the comment of Referee #1. (3) The author receives seriously the referee's comment, and generally agrees with it. He added new Figures 4 and 5, and Table 4 in order to improve the accuracy of the simulated results as much as possible, as described in the above.

The author added the above (1)-(3) explanations in the revised manuscript.

Please also note the supplement to this comment:
https://www.biogeosciences-discuss.net/bg-2017-447/bg-2017-447-AC2-supplement.zip

———————————————